# Evaluation of Mandibular Bone Alterations by Panoramic Radiography: A Potential Tool in the Identification of Signs of Osteopenia and Osteoporosis

**DOI:** 10.3390/bioengineering12070746

**Published:** 2025-07-09

**Authors:** Esdras Gabriel Alves-Silva, Betania Fachetti Ribeiro, Camila Fontes Silva, Rita de Kássia-Alves, Rodrigo Arruda-Vasconcelos, Lidiane Mendes Louzada, Rebecca F. Almeida-Gomes, João Miguel Marques Santos, Brenda P. F. A. Gomes

**Affiliations:** 1Department of Restorative Dentistry, Division of Endodontics, Piracicaba Dental School, University of Campinas—UNICAMP, Piracicaba 13083-970, SP, Brazil; esdras0702@yahoo.com.br (E.G.A.-S.); or ravasconcelos@usp.br (R.A.-V.); lidiane.mlouzada@gmail.com (L.M.L.); 2Department of Dentistry, University Federal of Rio Grande do Norte—UFRN, Natal 59056-000, RN, Brazil; betaniafachetti@hotmail.com; 3Division of Dentistry, Potiguar University—UnP, Natal 59056-000, RN, Brazil; camilafontessilva4@gmail.com (C.F.S.); riiittaalves@hotmail.com (R.d.K.-A.); 4Department of Dentistry, School of Dentistry, University of São Paulo, São Paulo 05508-000, SP, Brazil; 5Faculty of Medical Sciences, University Center Lusíada—UNILUS, Santos 11050-071, SP, Brazil; rebeccaagomes@outlook.com; 6Institute of Endodontics, Faculty of Medicine, University of Coimbra, 3000-075 Coimbra, Portugal

**Keywords:** osteoporosis, early diagnosis, panoramic radiograph, dentistry, radiology

## Abstract

This study aimed to evaluate the validity of panoramic radiography as an auxiliary method for identifying mandibular bone features consistent with a diagnosis of osteopenia or osteoporosis. Ninety panoramic radiographs were analyzed to assess the quality of the mandibular cortical layer below the mental foramen on both sides of the mandible. Scores C1 (normal), C2 (osteopenia), and C3 (osteoporosis) were attributed according to the cortical morphology. The sample consisted of 78 (86%) women aged 45 years or older and 12 (14%) men older than 60 years old. In 39 (43%) cases, the C1 score was evidenced as the lower mandibular cortical layer was normal on the image. The C2 score was identified in 47 (52%) cases, in which the cortical layer showed semilunar defects. Four (5%) cases presented a C3 score, with the cortical layer showing a clearly porous, thinner bone cortex. The presence of risk behaviors (e.g., smoking and alcoholism) as well as some comorbidities (e.g., systemic arterial hypertension, diabetes mellitus and thyroid disorders) was also observed. Mandibular bone changes were observed in association with a set of risk factors using panoramic radiography.

## 1. Introduction

Osteoporosis (OTP) is a systemic disease that has emerged as a serious public health problem due to its high morbidity rates and high social costs [1,2,3,4,5]. Approximately 10 million people in Brazil are affected by osteoporosis, yet only 20% are aware of their condition. This lack of awareness contributes to a staggering 200,000 deaths annually. Often called the “silent disease,” osteoporosis typically remains asymptomatic until its most serious complication, fractures, occurs [6]. This skeletal disorder is characterized by bone fragility caused by deterioration of the bone micro-architecture, which imparts strength and bone quality. The progressive loss of bone mass and structural degradation increase bone fragility and susceptibility to fractures, even from minor trauma, and are associated with severe complications, including a risk of death [3,7,8]. Bone mineral density can be affected in various parts of the body, leading to hollows and fractures, particularly in the vertebrae, femur, forearm, and jaw bones. In the oral cavity, the disease’s main consequences are alveolar ridge resorption and tooth loss, which can contribute to numerous oral health problems and decrease self-esteem for many individuals [4,9].

The skeleton maintains bone homeostasis through a finely tuned balance between bone formation and resorption [10]. In osteoporosis, this equilibrium is disrupted, resulting in increased bone turnover and impaired bone remodeling, which alter the bone matrix and mineral composition—hallmark features of the disease [10]. Bone mesenchymal stem cells (MSCs) are essential to the bone formation process and also play crucial roles in orofacial bone modeling and remodeling. Notably, orofacial MSCs (OFMSCs) differ phenotypically and functionally from MSCs found in non-oral skeletal sites [11]. OFMSCs exhibit high proliferative capacity and significantly greater population-doubling potential. Age-related skeletal changes are also evident in the jaws, with craniofacial bones—particularly the mandible—commonly displaying radiological signs of osteoporosis [12,13]. The mandible’s distinct cortical rim is prone to microarchitectural deterioration, including increased cortical porosity and reduced cortical thickness, due to heightened bone remodeling and a shift toward negative remodeling balance. Both the maxilla and mandible also exhibit thinning and loss of trabecular bone, further reflecting the systemic impacts of osteoporosis [12].

The disease can affect bone mineral density in various parts of the body, causing hollows and fractures, particularly vertebrae, femur and forearm bones, as well as jaw bones. In the oral cavity, the main consequences are alveolar ridge resorption and tooth loss, among other occurrences, which can cause various oral health problems and low self-esteem in thousands of people [4,8].

The disease’s progression is influenced by several risk factors, including gender (with a higher prevalence in women), advancing age, ethnicity (especially in white and Asian populations), and lifestyle factors such as low calcium and vitamin D intake, in-adequate sun exposure, physical inactivity, excessive caffeine consumption, smoking, and alcohol abuse [4,6]. Although osteoporosis affects both genders, post-menopausal women over 65 years old are at the greatest risk due to the sharp decline in estrogen levels that accelerates bone loss. Men, particularly those over 70, are also vulnerable, though the onset tends to occur later in life. If not diagnosed and managed early, the disease can severely impair the quality of life and result in fatal complications from fall-related fractures. Therefore, the identification of individuals at high risk for osteoporotic fractures is the basis of any preventive program [5,7,14].

Osteoporosis and osteopenia represent a spectrum of skeletal conditions characterized by reduced bone density and structural deterioration of bone tissue. Osteopenia, often considered a precursor to osteoporosis, indicates bone density that is lower than normal but not low enough to be classified as osteoporosis. While osteopenia itself may not always result in immediate health problems, it serves as a critical warning sign of potential progression to osteoporosis, a more severe condition associated with high morbidity and significant social and economic costs [14].

While dual-energy X-ray absorptiometry (DXA) is currently the gold standard for diagnosing osteoporosis, its high cost and limited accessibility, especially in low-resource settings, hinder its widespread use [1,8,15]. Consequently, alternative diagnostic methods have been explored. Panoramic radiography (PR), a commonly used imaging technique in dental practice, presents a more affordable and accessible option for assessing osteoporosis risk. Studies suggest that PR, particularly when combined with the Klemetti classification of mandibular bone morphology, could serve as a valuable tool for evaluating bone quality in the jaw and screening for osteoporosis in broader populations [3,15,16,17].

Lower mandibular cortex thickness and trabecular pattern observed on panoramic radiographs are useful tools for identifying individuals at risk of developing osteoporosis [10]. Although these findings are supported by studies evaluating the panoramic mandibular index alongside DXA measurements [17,18], there remains a significant gap in the literature concerning risk factors—especially within the Brazilian population—for patients diagnosed with osteopenia and osteoporosis, as well as the validation of Klemetti’s method [18].

The present study aimed to evaluate the validity of panoramic radiography for assessing mandibular bone quality using the Klemetti classification, with a specific focus on its potential role in identifying bone conditions consistent with osteopenia or osteoporosis in the lower jaw.

The null hypothesis of this study asserts that panoramic radiography is ineffective for assessing mandibular bone quality in the early diagnosis of osteoporosis.

## 2. Materials and Methods

This study was approved by the Ethics and Research Committee, according to registration number CAAE 56191716.7.0000.5296. This was an observational, retrospective, descriptive study, developed from March to November 2018. Patients treated at Universidade Potiguar (UnP) were selected due to the need for panoramic radiography, with fully completed clinical records and informed consent forms obtained prior to participation in the study. Given the retrospective design, anonymized data exclusively were used, ensuring that no personally identifiable information was accessible during the analysis. Patient confidentiality was rigorously maintained in compliance with data protection regulations.

Sample size calculation, with 85% power (β = 0.15), indicated a minimum sample size of 90 panoramic radiographs to show a 5% (α = 0.05). The calculations were made in the GPower 3.1 (Heinrich Heine University, Dusseldorf, Germany).

Initially, 300 clinical records of patients seeking dental treatment at the Dental School were analyzed. Of these, 52 patients did not fit the age group, 78 had incomplete clinical records (which allowed no evaluation of the systemic involvement), 32 had clinical records showing the presence of renal diseases or cancer involvement, and 48 had inadequate panoramic radiographs. In the end, a total of 90 patients were included, which is shown in Figure 1.

The inclusion criteria were as follows: panoramic radiographs of women aged 45 or older and men aged 60 or older; no previous diagnosis of OTP; standard radiographic images acceptable for analysis; complete medical records with a signed informed consent form.

The exclusion criteria were as follows: age range outside the study limits; incomplete medical records; use of drugs that could affect bone metabolism; inadequate radiographic images with high or low radiographic density; and the presence of radiographic findings that could interfere with the analysis, such as osteitis, osteosclerosis, local destructive lesions and overlapping anatomical structures.

The PRs were performed by using a digital radiography device (Scanora model, Soredex, Tuusula, Uusimaa, Finland) operating at 77 kv, 8mA, and 15 s exposure time, according to the manufacturer’s specifications.

The panoramic images were analyzed with Radiocef Studio version 3—Radioimp computer software (Radiomemory, Belo Horizonte, MG, Brazil). The assessment was based on the mandibular cortical index [13], focusing on the quality of the mandibular cortical layer below the mental foramen on both sides of the mandible on panoramic radiographs. Classifications (C1, C2, C3) were attributed according to the cortical layer morphology, and their graphical presentations can be seen in Table 1 and Figure 2.

To validate the method, 20 panoramic radiographies of female patients aged 45 years or older without osteopenia/osteoporosis diagnosis and 20 panoramic radiographies of patients with osteoporosis/osteopenia diagnosed on at least two of the following three different sites analyzed (lumbar spine, femur, or middle third of the radius) were selected. These 20 patients with confirmed osteoporosis were not included in the study sample but were used to validate the Klemetti method [18].

Osteopenia/osteoporosis diagnosis was based on bone densitometry exam (DEXA—dual energy X-ray absorptiometry). This criterion is used to define the osteoporosis diagnosis according to the World Health Organization. Values are: Normal = scores up to −1 standard deviation (SD) from the young female adult mean; Osteopenia = score between −1.1 and −2.5 SD below the mean; and Osteoporosis = score of −2.6 SD or more below the mean. Of these patients, ten were women aged 65 years or older and ten were men aged 70 years or older. These images were also used to compare radiographic signs suggestive of OTP on the radiographs of patients without a previous diagnosis of the disease. It could also help to correlate these signals with injury levels established elsewhere [13].

### Statistical Analysis

The data observed on the panoramic radiographies, together with gender, age, and medical history listed on the spreadsheet, were analyzed by using SPSS, version number 31 for Windows (SPSS Inc., Chicago, IL, USA). Absolute and relative frequencies and the 95% confidence interval were calculated.

The panoramic images were analyzed using the mandibular cortical index (Klemetti [18]) to assess the quality of the mandibular cortical layer below the mental foramen on both sides of the mandible. Classifications (C1, C2, C3) were assigned based on the cortical quality.

The panoramic images were assessed twice in random order by double-blind examiners at a 1-week interval without knowing the previous results. In case of disagreement, the images were jointly re-evaluated by both examiners.

The weighted coefficient kappa (Kw) was used to measure the inter-rater reproducibility between the examiners separately for each time period and to measure the intra-rater reproducibility between the time periods separately for each examiner. The reliability analysis revealed a Kappa between 0.68 and 0.90 with a 95% confidence interval, indicating good agreement.

## 3. Results

Out of the 90 patients included in this research, 78 (86%) were women aged 45 years or older, and 12 (14%) were men aged 60 years or older. Among the women, their ages ranged from 45 to 72 years, with a mean age of 56 years. Among the men, their ages ranged from 60 to 89 years, with a mean age of 68 years (Table 2). Table 3 shows the distribution of the 90 PRs analyzed according to the mandibular cortical layer. The C1 score was attributed to 39 (43%) radiographs in which the mandibular cortical layer was normal, smooth, and regular (Figure 2B). In 47 (52%) cases, semilunar defects were identified in the cortical bone, resulting in a C2 score (Figure 2C). Four (5%) panoramic radiographs showed that the cortical mandibular layer was clearly porous and thinner, resulting in a C3 score (Figure 2C).

With regard to age and gender, it was observed that women between 45 and 50 years old (*n* = 33), even in the pre-menopausal phase, had a normal cortical layer. Cortical changes indicative of osteopenia (C2) were detected in individuals aged 50–60 years old (*n* = 43). Only two 72-year-old women presented with osteoporosis.

Among the males, six patients aged 60 years old had normal cortical layer, four aged 65 years old presented with osteopenia, and two aged 70 years old had osteoporosis.

With regard to the possible etiological factors associated with the disease, it was observed that their clinical records showed some clinical risk factors, such as smoking and alcoholism, as well as some comorbidities (e.g., systemic arterial hypertension and diabetes mellitus). In addition, thyroid disorders were also observed (Table 4).

## 4. Discussion

To date, there is no accurate evaluation of the average bone strength. Bone mineral density (BMD) is used as an approximate average because it contributes approximately to about 70% of bone strength [14]. The World Health Organization (WHO) defines osteoporosis based on BMD values, with a BMD index more than 2.5 standard deviations (SD) below the average for healthy young white women [14,19,20].

In clinical practice, a T-score is used to classify bone density. A T-score of up to −1 SD is considered normal, while a score between −1 and −2.5 SD indicates osteopenia, a condition where bone mass is reduced but not yet severe enough to be classified as osteoporosis. A T-score of −2.5 SD or lower is indicative of established osteoporosis, often accompanied by fragility fractures. These criteria apply to measurements taken from key skeletal sites, including the lumbar spine, femur, and the middle third of the radius [14,21].

Reproducibility, which indicates agreement between repeated measurements by observers, showed that there was reproducibility between the first and second observer for calibration indices in our study, which was in agreement with the results of the study conducted by Alapati et al. [1] and Klementti et al. [18], which was attributed to the higher level of understanding of the descriptive analysis pattern by the first and second observers.

Although osteoporosis is more prevalent among women, especially those over 65 years old, this study included both male and female participants. While women are disproportionately affected by the disease, particularly after menopause, the inclusion of men was necessary given that osteoporosis is also prevalent in older males, with diagnoses often delayed compared to females [3,8,14].

In this study, 51 out of 90 patients (57%) exhibited some degree of bone loss in the mandibular cortical layer on panoramic radiographs (PRs), consistent with findings in other studies [9]. Osteoporosis predominantly affects bones such as the femur, wrist, and spine, particularly in women, while osteopenia, characterized by gradual bone mass loss, serves as a precursor to osteoporosis. Factors contributing to osteopenia include aging, which increases bone porosity and hinders calcium absorption, as well as genetic predisposition, malnutrition, insufficient sun exposure, and physical inactivity [2,4,8].

In our study, women aged 45 to 50 exhibited normal PR findings. However, after the age of 55 and in the menopausal phase, a predominance of osteopenia was observed, aligning with previous reports [4,7,17]. This decline in bone mass is likely linked to reduced estrogen production, a hormone crucial for calcium absorption and bone metabolism. Women experiencing early menopause, particularly those of low body weight and height, are especially vulnerable to this loss of bone density [1]. In men, osteopenia was more commonly observed after the age of 70, when estrogen levels also begin to decline [4,17,22,23,24].

The results rejected the null hypothesis, as panoramic radiography was shown to be effective in adjunctive assessment of mandibular bone quality in the early diagnosis of osteoporosis.

Dagistan et al. [25] identified a significant association between the mandibular cortical index (MCI) and osteoporosis, aligning with the findings of our study. These results underscore the growing importance of MCI as a reliable tool for identifying at-risk groups [6]. Our research demonstrated that the thickness and shape of the mandibular cortex, particularly the C2 and C3 patterns, appear to be associated with individuals at risk of developing osteoporosis. This reflects systemic bone mass loss and is closely aligned with the findings of Peycheva et al. [26], further validating the use of mandibular cortex assessments as an indicator of systemic bone health

Our findings indicate that osteopenia was more prevalent than osteoporosis within our sample, which comprised patients from a specific region in Brazil. This distribution may be influenced by demographic or regional characteristics, as well as lifestyle factors unique to this population, which likely impact bone density patterns [14]. It is important to note, however, that these results do not suggest osteoporosis is uncommon overall; rather, its lower prevalence reflects the characteristics of this specific sample.

Our data also showed that risk factors such as smoking, alcoholism, systemic arterial hypertension, diabetes mellitus, and thyroid dysfunction were frequently present in patient records. These factors have been identified as contributing to the onset of osteoporosis in other studies [4,7,27]. Hypertension, the most prevalent condition among individuals in the C2 group, may contribute to bone mass loss through excessive sodium and calcium excretion [4]. Additionally, antihypertensive medications, especially diuretics, can exacerbate calcium loss, further increasing the risk of osteoporosis [4,5,28].

There is also growing evidence of a link between diabetes mellitus and reduced bone mineral apposition rates, with diabetes being more common among osteopenic patients in this study, as noted in other research [4,21,29,30]. The association between diabetes and poor bone metabolism is likely due to decreased insulin-like growth factor-1 (IGF-1), which plays a key role in bone formation. This reduction in IGF-1 contributes to the onset of osteoporosis in diabetic patients.

Thyroid dysfunction, particularly hyperthyroidism, was also found to be a significant risk factor for osteoporosis, particularly in postmenopausal women. Thyroid hormones, particularly T3 and T4, have been shown to influence bone metabolism, and hyperthyroidism is associated with reduced bone mineral density in women over 65 years old [16,17,31,32,33].

Our findings underscore the importance of raising awareness about osteoporosis prevention, especially among postmenopausal women, who are at the highest risk. Early screening through panoramic radiography and bone densitometry is recommended to establish a baseline for future monitoring and treatment [14,21]. Preventive measures, including a balanced diet rich in calcium and vitamin D, adequate sun exposure, regular weight-bearing exercise, and avoidance of excessive alcohol and tobacco use, can significantly improve bone health [4,5,33].

In light of these findings, panoramic radiography can be regarded as a valuable adjunct in identifying early signs of systemic conditions like osteoporosis [1,2,3,5,7]. However, the search for other panoramic radiographic signs of bone loss associated with a quantitative index, such as fractal dimension (i.e., quantitative information about bone volume, density, and bone loss), should be encouraged. Detailed anamnesis should also be stimulated, including not only clinical data but also factors that may be associated with osteoporosis, such as physical inactivity, diet, and family history of diseases [7,14,17].

There is compelling evidence in the literature that dental panoramic radiography serves as an effective ancillary tool for assessing the risk of osteoporosis in postmenopausal women and elderly men. Early diagnosis or detection is crucial for initiating timely and successful therapy, making risk-assessment tools, including panoramic radiographs, essential in managing patients with reduced bone mineral density and at risk of vertebral and hip fractures [6].

Panoramic radiography is well-established as a complementary diagnostic method in dentistry. This study adds value by confirming the applicability of the Klemetti method [18] in a specific region of Brazil and highlighting the importance of identifying risk factors for osteopenia and osteoporosis. However, a notable limitation of this study is its small sample size, which suggests the need for future research with larger, more diverse populations. Expanding the sample size and comparing findings across different regions of Brazil and among varied patient groups will provide more comprehensive insights.

An additional limitation of the study lies in the sample selection process, which did not account for patients’ masticatory load, a factor that may significantly influence mandibular bone structure. This consideration warrants further investigation in future studies.

Further research is necessary to enhance the assessment of mandibular bone quality using panoramic radiography by integrating recent technologies. For instance, convolutional neural networks (CNNs) have been developed for the detection and enumeration of primary and permanent teeth in pediatric dental radiographs [34,35]. Additionally, smartphone applications are being explored to improve diagnostic capabilities. These advancements aim to improve the reliability of panoramic radiography as an auxiliary method for the early diagnosis of osteoporosis in daily clinical practice.

Given the routine use of panoramic radiography in dental practice, dental surgeons can play a crucial role in the early detection of osteoporosis. When signs of osteoporosis are observed on a panoramic radiograph, patients should be informed and referred for further evaluation, including bone densitometry, to confirm the diagnosis and initiate appropriate treatment [15,16,17,36]. Although radiographic changes in the mandibular cortical bone may not significantly improve with osteoporosis treatment, early diagnosis remains essential for effective management. Finally, multicenter clinical trials with larger sample sizes are needed to improve diagnostic, therapeutic, and preventive strategies and to deepen our understanding of osteoporosis etiology.

## 5. Conclusions

Within the limitations of this study, it can be concluded that changes in the mandibular cortical bone observed in the panoramic radiographs are associated with sistemic risk factors for osteoporosis. Given the regular use of panoramic radiographs, dental professionals can play a key role in early detection and referral.

## Figures and Tables

**Figure 1 bioengineering-12-00746-f001:**
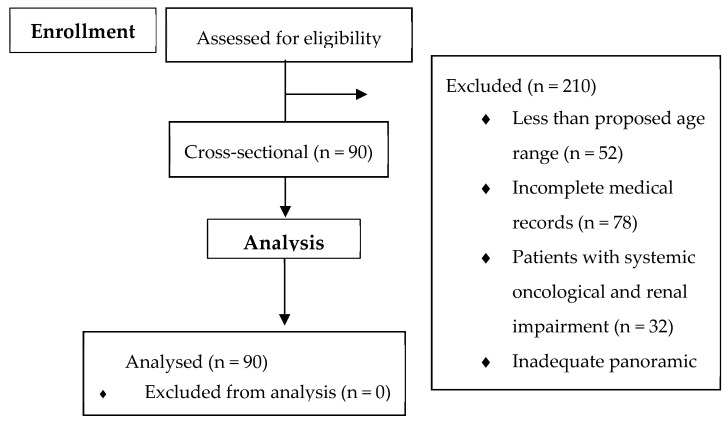
Flow diagram of the study.

**Figure 2 bioengineering-12-00746-f002:**
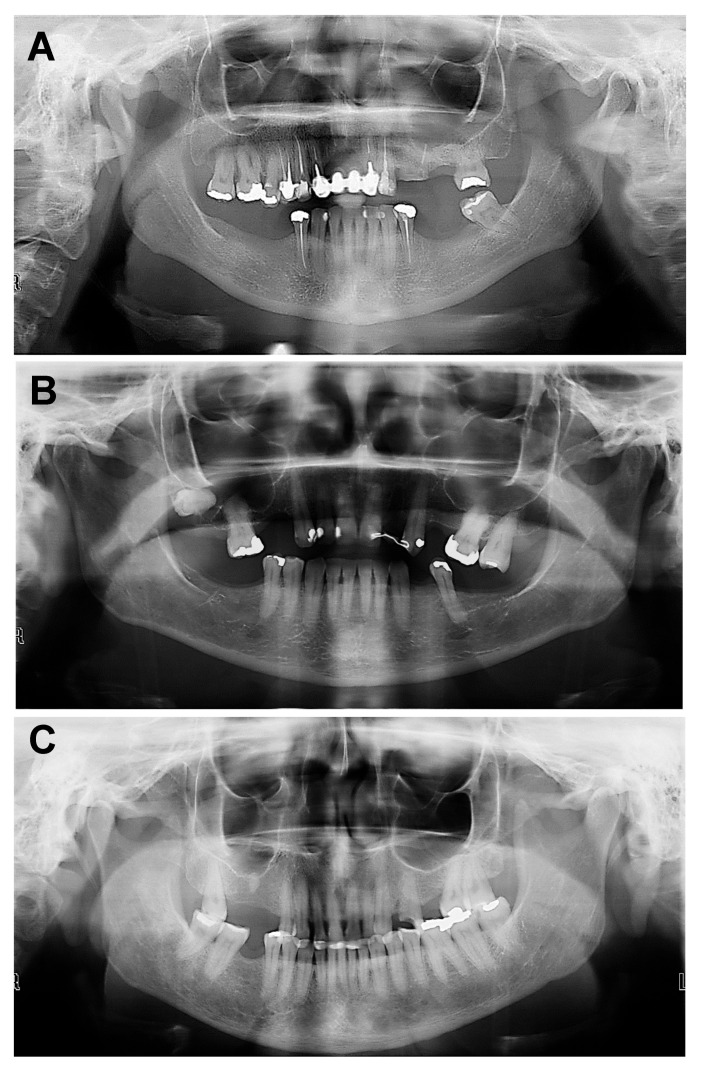
(**A**) C1—Smooth and regular cortical; (**B**) C2—Cortical showing semilunar defects (lacunar resorption); (**C**) C3—Cortical presenting porous and with erosion.

**Table 1 bioengineering-12-00746-t001:** Classification by cortical layer.

Classification	Definition
C1	(Normal)—Smooth and regular cortical
C2	(Osteopenia)—Cortical with semilunar defects (lacunar resorption)
C3	(Osteoporosis)—Clearly porous cortical layer and its thickness is tapered and reduced

**Table 2 bioengineering-12-00746-t002:** Frequency, mean, and standard deviation of age according to gender.

Gender	*n* (%)	Mean (Standard Deviation)
Women	78 (86%)	57.34 (8.56)
Male	12 (14%)	68.50 (10.77)
Total	90 (100%)	58.86 (9.57)

**Table 3 bioengineering-12-00746-t003:** Distribution of panoramic radiographs according to the classification of the lower mandible cortex.

Score	Total	Women	Male
	*n* (%)	*n* (%)	*n* (%)
C1	39 (43)	33 (84)	6 (16)
C2	47 (52)	43 (91)	4 (9)
C3	4 (5)	2 (50)	2 (50)
Total	90 (100)	78 (86)	12 (14)

C1 Normal, C2 Osteopenia, C3 Osteoporosis.

**Table 4 bioengineering-12-00746-t004:** Aetiological factors associated with lower mandible cortical classification.

Score	TB	ET	SAH	DM	TD
**C1**	6	1	5	8	2
**C2**	3	0	18	9	0
**C3**	0	0	1	0	1

C1 Normal, C2 Osteopenia, C3 Osteoporosis; Smoking (TB); Ethylism (ET); Systemic arterial hypertension (SAH); Diabetes Mellitus (DM); Thyroid Disorders (TD).

## Data Availability

Data will be made available upon request to the corresponding authors.

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
