# Peer review of "Evaluation of Mandibular Bone Alterations by Panoramic Radiography: A Potential Tool in the Identification of Signs of Osteopenia and Osteoporosis"

_bioengineering, 2025, doi:10.3390/bioengineering12070746_

Round 1

Reviewer 1 Report

Comments and Suggestions for Authors

Thank you for giving me the opportunity to review this paper on the auxiliary use of panoramic radiography for the early diagnosis of osteoporosis. Early detection of osteoporosis, a silent disease but one that greatly compromises patients' quality of life, is a very important area of research. The story of how dental examination can contribute to the early detection of osteoporosis is very exciting. However, although this is an auxiliary diagnostic method for osteoporosis, rigorous verification of its effectiveness is required. Additional explanations are necessary, especially regarding the research design. The paper would be more robust if the following points were added to the discussion.

Introduction

P2L47-51, P2L68-71

The background of osteoporosis, diagnostic methods, and their relevance to dentistry are described in detail, making it easy for the reader to understand. However, it seems that similar sentences are repeated. To simplify the author's argument, one of them should be deleted.

P3L106-

Previous studies have shown that MCI is effective in diagnosing osteoporosis. The issues raised by the author have been resolved. The authors should endeavor to elucidate additional issues in this study.

Methods

P3L113-115

The research design should be described in detail. Did the authors plan a cross-sectional study?

P3L127-131

Why exclude osteoporosis and anti-resorptive agent users? Do the authors only want to conduct a cross-sectional study of healthy individuals? If that logic is applied, the author's stated purpose of verifying the effectiveness of imaging diagnoses of osteoporosis patients using PR images cannot be achieved. The author should explain this paradox.

P4 Figure1

As mentioned above, the authors should clarify the criteria for validating their own objectives and modify Figure 1.

P4 Table1

Previous studies have not reported diagnostic criteria that C2 is osteopenia and C3 is osteoporosis. Statements for which no evidence has been obtained should not be made.

P5 Figure 2

It is preferable to present a sample of C1-C3, but it is too small to discern MCI at all.

A larger figure should be used to solve this problem.

Results

P7 Table4

This table only provides a descriptive summary and does not analyze pathogenic factors. To show the results of factor analysis, regression analysis must be performed and odds ratios must be presented. Can't the correlation be shown by the results of this study?

Discussion

P7L265

It is unclear what the number 1 means.

P8L271-273

This result cannot be determined from the findings of this study. It should be considered speculative.

P8L273-275

Using MCI to diagnose osteoporosis is an excellent discovery. We hope this field of research will grow in the future. However, more research is needed to understand the trends that MCI might show in patients with osteoporosis and in those already using ARA. The authors should consider the following studies. The following studies should be cited to add to the author's discussion.

Seki K, Nagasaki M, Yoshino T, Yano M, Kawamoto A, Shimizu O, Radiographical Diagnostic Evaluation of Mandibular Cortical Index Classification and Mandibular Cortical Width in Female Patients Prescribed Antiosteoporosis Medication: A Retrospective Cohort Study. Diagnostics 14, 1009

P8L287

It is unclear what the number 4 means.

P9L334

In the discussion, it is necessary to state whether the hypothesis stated by the author in the opening paragraph has been rejected.

Author Response

# Author's Reply to the Review Report (Reviewer 1).

Thank you for giving me the opportunity to review this paper on the auxiliary use of panoramic radiography for the early diagnosis of osteoporosis. Early detection of osteoporosis, a silent disease but one that greatly compromises patients' quality of life, is a very important area of research. The story of how dental examination can contribute to the early detection of osteoporosis is very exciting. However, although this is an auxiliary diagnostic method for osteoporosis, rigorous verification of its effectiveness is required. Additional explanations are necessary, especially regarding the research design. The paper would be more robust if the following points were added to the discussion.

Our response: We appreciate the observation of reviewer 1 and highlight the importance of using panoramic imaging as an auxiliary method in the diagnosis of osteoporosis.

1.Introduction: P2L47-51, P2L68-71 - The background of osteoporosis, diagnostic methods, and their relevance to dentistry are described in detail, making it easy for the reader to understand. However, it seems that similar sentences are repeated. To simplify the author's argument, one of them should be deleted.

Our response: We appreciate the observation made by reviewer 1 and emphasize that the text in P2L68-71 has been removed.

  1. Introduction: P3L106 - Previous studies have shown that MCI is effective in diagnosing osteoporosis. The issues raised by the author have been resolved. The authors should endeavor to elucidate additional issues in this study.

Our response: We appreciate this insightful observation and acknowledge the concerns raised regarding the limited information provided on the risks associated with osteopenia and osteoporosis. We also recognize the importance of addressing the validation and reliability of the Klemetti method (1994). In response, we have revised the manuscript to include a more comprehensive discussion of these topics.

  1. Methods: P3L113-115 - The research design should be described in detail. Did the authors plan a cross-sectional study?

Our response: We appreciate the observation made by reviewer 1 and emphasize that the research design should be better described to improve the reader's understanding and that a cross-sectional study is also in the group's interest.

  1. Methods: P3L127-131 - Why exclude osteoporosis and anti-resorptive agent users? Do the authors only want to conduct a cross-sectional study of healthy individuals? If that logic is applied, the author's stated purpose of verifying the effectiveness of imaging diagnoses of osteoporosis patients using PR images cannot be achieved. The author should explain this paradox.

Our response: We appreciate the careful consideration and comments provided by Reviewer 1. We would like to clarify that the initial exclusion of patients diagnosed with osteoporosis was necessary to validate the Klemetti method by assessing whether individuals without a prior diagnosis might still be predisposed to the condition. Subsequently, panoramic radiographs of patients with confirmed osteoporosis were included to correlate the radiographic findings between patients with and without the diagnosis.

  1. P4 Figure1 - As mentioned above, the authors should clarify the criteria for validating their own objectives and modify Figure 1.

Our response: We appreciate the observation made by reviewer 1 and highlight that in P4L145-149 it is presented: To validate the method, 20 panoramic radiographs of female patients aged 45 years or older without a diagnosis of osteopenia/osteoporosis and 20 panoramic radiographs of patients diagnosed with osteoporosis/osteopenia in at least two of the three sites analyzed (lumbar spine, femur or middle third of the radius) were selected. These 20 patients with confirmed osteoporosis were not included in the study sample but were used to validate the Klemetti method [18].

  1. P4 Table1 - Previous studies have not reported diagnostic criteria that C2 is osteopenia and C3 is osteoporosis. Statements for which no evidence has been obtained should not be made.

Our response: We appreciate the observation made by Reviewer 1 and would like to emphasize that, according to the literature, the C2 classification is associated with osteopenia and C3 with osteoporosis, particularly when compared with patients diagnosed using DEXA. This correlation is supported by recent findings, such as those presented in the following article: https://doi.org/10.1007/s11914-023-00807-5.

P5 Figure 2 - It is preferable to present a sample of C1-C3, but it is too small to discern MCI at all. A larger figure should be used to solve this problem.

Our response: We appreciate the suggestion made by reviewer 1 and note that the larger forms to enhance the visualization of the MCI.

  1. Results: P7 Table4 - This table only provides a descriptive summary and does not analyze pathogenic factors. To show the results of factor analysis, regression analysis must be performed and odds ratios must be presented. Can't the correlation be shown by the results of this study?

Our response: We appreciate the observation made by reviewer 1 and emphasize that it was not possible to perform a regression analysis. We suggest this for future work.

  1. Discussion: P7L265 - It is unclear what the number 1 means.

Our Response: We are grateful to Reviewer 1 for pointing this out and acknowledge that the reference contained a typographical mistake. 

  1. Discussion: P8L271-273 - This result cannot be determined from the findings of this study. It should be considered speculative.

Our response: We appreciate the observation made by reviewer 1 and have amended the text as follows: Our research demonstrated that the thickness and shape of the mandibular cortex, particularly the C2 and C3 patterns, appear to be associated with individuals at risk of developing osteoporosis.

  1. Discussion: P8L273-275 - Using MCI to diagnose osteoporosis is an excellent discovery. We hope this field of research will grow in the future. However, more research is needed to understand the trends that MCI might show in patients with osteoporosis and in those already using ARA. The authors should consider the following studies. The following studies should be cited to add to the author's discussion.

Seki K, Nagasaki M, Yoshino T, Yano M, Kawamoto A, Shimizu O, Radiographical Diagnostic Evaluation of Mandibular Cortical Index Classification and Mandibular Cortical Width in Female Patients Prescribed Antiosteoporosis Medication: A Retrospective Cohort Study. Diagnostics 14, 1009

Our response: We agree with the suggestion of reviewer 1 and emphasize that the valuable article was included in the discussion.

  1. Discussion: P8L287 - It is unclear what the number 4 means.

Our Response: We are grateful to Reviewer 1 for pointing this out and acknowledge that the reference contained a typographical mistake. 

  1. Discussion: P9L334 - In the discussion, it is necessary to state whether the hypothesis stated by the author in the opening paragraph has been rejected.

Our response: We appreciate Reviewer 1’s suggestion and have emphasized in the text that the null hypothesis was rejected.

Reviewer 2 Report

Comments and Suggestions for Authors

Gomes and colleagues reported that panoramic radiography has demonstrated significant potential as a valuable tool for the early detection of reduced mandibular bone quality, which may be indicative of systemic bone loss. In their study, 57% of cases exhibited cortical alterations consistent with osteopenia or osteoporosis. These findings suggest that panoramic radiographs—commonly employed in routine dental evaluations—can serve as an effective preliminary screening method for identifying patients at risk of osteoporosis, particularly when combined with clinical risk factors such as age, smoking, alcohol use, and comorbid conditions including hypertension, diabetes, and thyroid disorders. Overall, this work appears suitable for publication in Bioengineering, pending minor revisions.

  1. Please ensure that all references in the manuscript are rewritten and formatted according to the Bioengineering journal guidelines for clarity and consistency.

Author Response

# Author's Reply to the Review Report (Reviewer 2).

Gomes and colleagues reported that panoramic radiography has demonstrated significant potential as a valuable tool for the early detection of reduced mandibular bone quality, which may be indicative of systemic bone loss. In their study, 57% of cases exhibited cortical alterations consistent with osteopenia or osteoporosis. These findings suggest that panoramic radiographs—commonly employed in routine dental evaluations—can serve as an effective preliminary screening method for identifying patients at risk of osteoporosis, particularly when combined with clinical risk factors such as age, smoking, alcohol use, and comorbid conditions including hypertension, diabetes, and thyroid disorders. Overall, this work appears suitable for publication in Bioengineering, pending minor revisions.

  1. Please ensure that all references in the manuscript are rewritten and formatted according to the Bioengineering journal guidelines for clarity and consistency.

Our response: We appreciate the observation made regarding the article and emphasize that all references were formatted according to the journal guidelines.

Reviewer 3 Report

Comments and Suggestions for Authors

The authors have conducted a study the use of panoramic radiography to detect early signs of osteopenic/osteoporotic alterations in a selected group of patients. The reviewer considers that the study has potential. Nonetheless the following questtions must be addressed:

  1. Introduction section. Lines 103-108 are not properly referenced. This must be corrected
  2. Introduction section. Lines 103-105 is not entirely accurated, as several risk factors for osteoporosis are identified. Authors can state that all risk factors are not yet characterized, or information regarding brazilian population is lacking. This shoulde be corrected accordingly.
  3. Materials and methods refers the acquisition of linear meausurements, however only the results for qualitative assessment are presented. This should be clarified.
  4. No data on the dentition quality, presence of posterior sector, pedriodontal status,... of the selected patients was considered for the analysis of the results. However mechanical mastiacatory load plays a very relevant role for  the maintenance of mandibular bone structure. Therefore, potential bias is introduced in the data collection. Did authors collect information in this regard? This should be corrected accordingly. If not, it has to be clearly stated in the discussion section, that it represents a significant limitation of the study.
  5. Discussion section fails to state in the first paragraphs if the attained data supports the initial study hypothesis. This should be corrected accordingly.
  6. The major concern of the reviewer relates with framing of the study. Klemeti method is a established score for assess bone quality, therefore is redundant to validate it with radiographs of osteoporotic/osteopenic patients, especially if no conclusive diagnosis of those conditions was established for the sample patients. In this sense, the attained data shows that some patients in the sample have alterations of bone quality detected in PR. However, authors have quite relevant data, regarding characterization of potential associated risk factors . The strenght of the study is precisely that association, that might help to outline a risk profile for the brazilian population, and improve diagnosis of bone altering conditions. Results section and iscussion section should be redirected to highlight those findings.

Author Response

# Author's Reply to the Review Report (Reviewer 3).

The authors have conducted a study the use of panoramic radiography to detect early signs of osteopenic/osteoporotic alterations in a selected group of patients. The reviewer considers that the study has potential. Nonetheless the following questtions must be addressed:

  1. Introduction section. Lines 103-108 are not properly referenced. This must be corrected

Our response: We appreciate the observation made by reviewer 3 and highlight that the references have been included in the text.

  1. Introduction section. Lines 103-105 is not entirely accurated, as several risk factors for osteoporosis are identified. Authors can state that all risk factors are not yet characterized, or information regarding brazilian population is lacking. This shoulde be corrected accordingly.

Our response: We appreciate Reviewer 3’s suggestion and have addressed the concern regarding risk factors in the Brazilian population within the manuscript.

  1. Materials and methods refers the acquisition of linear meausurements, however only the results for qualitative assessment are presented. This should be clarified.

Our response: We thank Reviewer 3 for the valuable observation and clarify that the present study has a retrospective design, focusing on the role of panoramic radiography as a complementary tool in the early detection of osteoporosis. Therefore, the data are qualitative in nature. The study involved sample selection and validation of the Klemetti method by examining panoramic radiographic data and assessing associated risk factors

  1. No data on the dentition quality, presence of posterior sector, pedriodontal status,... of the selected patients was considered for the analysis of the results. However mechanical mastiacatory load plays a very relevant role for  the maintenance of mandibular bone structure. Therefore, potential bias is introduced in the data collection. Did authors collect information in this regard? This should be corrected accordingly. If not, it has to be clearly stated in the discussion section, that it represents a significant limitation of the study.

Our response: We appreciate Reviewer 3’s careful observation and clarify that this study limitation has been included in the relevant section.

  1. Discussion section fails to state in the first paragraphs if the attained data supports the initial study hypothesis. This should be corrected accordingly.

Our response: We appreciate the suggestion made by reviewer 1 and emphasize that it was included in the text that the null hypothesis was rejected.

  1. The major concern of the reviewer relates with framing of the study. Klemeti method is a established score for assess bone quality, therefore is redundant to validate it with radiographs of osteoporotic/osteopenic patients, especially if no conclusive diagnosis of those conditions was established for the sample patients. In this sense, the attained data shows that some patients in the sample have alterations of bone quality detected in PR. However, authors have quite relevant data, regarding characterization of potential associated risk factors. The strenght of the study is precisely that association, that might help to outline a risk profile for the brazilian population, and improve diagnosis of bone altering conditions. Results section and iscussion section should be redirected to highlight those findings.

Our Response: We appreciate and understand the concern raised by Reviewer 3. As detailed in the methodology section, 20 panoramic radiographs (PRs) from patients with a prior diagnosis of osteopenia/osteoporosis and 20 PRs from patients without such a diagnosis were selected to validate the Klemetti method. However, it is important to clarify that the 90 PRs analyzed in the main study were from patients without a diagnosis of osteopenia or osteoporosis.

Round 2

Reviewer 1 Report

Comments and Suggestions for Authors

Congratulations!

All modifications confirmed.

Author Response

Comments and Suggestions for Authors: Congratulations!

All modifications confirmed.

Our response: We sincerely thank the reviewer for the positive feedback and for confirming all the modifications. We truly appreciate your time and valuable contributions to the improvement of our manuscript.

Reviewer 3 Report

Comments and Suggestions for Authors

Upon carefull analysis of the revised version, the reviewer regrets to inform that the manuscript does not fulfill the requirements to endorse its publication.

  1. Point 3 was not addressed by the authors. The revised version states that linear measurements (line139-140) were made, however only qualitative data is presented. therefore it is not clear to what authors refer.
  2. Point 6, the most crucial aspect, was not addressed by the authors. Authors have used the klemetti index to determine bone mandibular alterantions in selected group of patients. Although those alterations are coherent with osteoporotic/osteopenic alterations, it cannot be stated that the sample patients present any of the conditions, since no final diagnosis was conducted, therefore patients might be or not osteoporotic. This is aggravated by the lacking of information on dental condition. Authors can only state that mandibular bone alterations were detected combined with a set of risk factors. In this sense, the conclusion- PR detects early signs of osteoporosis-of the authors is not supported by the results, since it is unclear if the detected bone alteration are indeed caused by osteoporosis or other bone altering condition.  

Author Response

  1. Point 3 was not addressed by the authors. The revised version states that linear measurements (line139-140) were made, however only qualitative data is presented. Therefore it is not clear to what authors refer.

We appreciate the reviewer’s observation. In response, the linear measurements (lines 139–140) refer to the mandibular cortical thicknesses (C1, C2, and C3) as defined in the Klemetti classification. We would like to highlight that, in the methodology section (lines 145–149), 20 panoramic radiographs from patients diagnosed with osteoporosis—confirmed by at least two of the following three sites: lumbar spine, femur, or mid-radius—were analyzed to correlate mandibular cortical bone findings with those of patients without osteoporosis. This approach allowed us to identify data consistent with osteopenia or osteoporosis. Additionally, we revised the manuscript to clarify the aim of detecting findings consistent with osteopenia or osteoporosis.

  1. Point 6, the most crucial aspect, was not addressed by the authors. Authors have used the klemetti index to determine bone mandibular alterantions in selected group of patients. Although those alterations are coherent with osteoporotic/osteopenic alterations, it cannot be stated that the sample patients present any of the conditions, since no final diagnosis was conducted, therefore patients might be or not osteoporotic. This is aggravated by the lacking of information on dental condition. Authors can only state that mandibular bone alterations were detected combined with a set of risk factors. In this sense, the conclusion- PR detects early signs of osteoporosis-of the authors is not supported by the results, since it is unclear if the detected bone alteration are indeed caused by osteoporosis or other bone altering condition.  

Our response: 1.  We thank the reviewer for their thoughtful comments and note that the conclusion of the study has been revised accordingly.

  1. The objective has also been adjusted to emphasize the use of panoramic radiography in the evaluation of mandibular bone changes.
  2. The title has been modified based on the reviewer’s suggestion to include the role of panoramic radiography in the detection of mandibular bone changes.
  3. It is important to highlight that the present study proposes panoramic radiography as a potential method for identifying signs consistent with osteoporosis.

Round 3

Reviewer 3 Report

Comments and Suggestions for Authors

The reviwer coniders that the main concerns were properly addressed by the authors.

It would be more accurate to remove the term "linear measurements" as it is highly suggestive of quantitative data collection, which can be misleading.

Author Response

The reviewer considers that the main concerns were properly addressed by the authors.

It would be more accurate to remove the term "linear measurements" as it is highly suggestive of quantitative data collection, which can be misleading.

Our response: We sincerely thank the reviewer for the positive feedback and performed the asked modification. We truly appreciate your time and valuable contributions to the improvement of our manuscript.